## PROCEEDINGS A

mechanics, mechanical engineering, applied mathematics

Gaussian curvature, active materials, metric compatibility, interface

**Author for correspondence:**
John S. Biggins
e-mail: jsb56@cam.ac.uk

†Deceased.

# Interfacial metric mechanics: stitching patterns of shape change in active sheets

Fan Feng[1], Daniel Duffy[1], Mark Warner[2,†] and John S. Biggins[1]

[1]Department of Engineering, University of Cambridge, Cambridge CB2 1PZ, UK
[2]Department of Physics, University of Cambridge, Cambridge CB3 0HE, UK

FF, 0000-0002-5456-670X; DD, 0000-0002-0383-5527; MW, 0000-0003-3172-0265; JSB, 0000-0002-7452-2421

A flat sheet programmed with a planar pattern of spontaneous shape change will morph into a curved surface. Such metric mechanics is seen in growing biological sheets, and may be engineered in actuating soft matter sheets such as phase-changing liquid crystal elastomers (LCEs), swelling gels and inflating baromorphs. Here, we show how to combine multiple patterns in a sheet by stitching regions of different shape changes together piecewise along interfaces. This approach allows simple patterns to be used as building blocks, and enables the design of multi-material or active/passive sheets. We give a general condition for an interface to be geometrically compatible, and explore its consequences for LCE/LCE, gel/gel and active/passive interfaces. In contraction/elongation systems such as LCEs, we find an infinite set of compatible interfaces between any pair of patterns along which the metric is discontinuous, and a finite number across which the metric is continuous. As an example, we find all possible interfaces between pairs of LCE logarithmic spiral patterns. By contrast, in isotropic systems such as swelling gels, only a finite number of continuous interfaces are available, greatly limiting the potential of stitching. In both continuous and discontinuous cases, we find the stitched interfaces generically carry singular Gaussian curvature, leading to intrinsically curved folds in the actuated surface. We give a general expression for the

distribution of this curvature, and a more specialized form for interfaces in LCE patterns. The interfaces thus also have rich geometric and mechanical properties in their own right.

## 1. Introduction

Sheets, plates and shells are traditionally used as stiff, passive structural elements. However, in recent years, the subject has been enlivened by the use of soft active materials to make shape-shifting elements [1–3] that recall the exquisite dynamism of biological tissues. These soft active materials typically undergo a large but homogeneous shape change on actuation: for example a gel dilates isotropically on swelling [4] and a liquid crystal elastomer (LCE) contracts uniaxially on heating or illumination [5]. However, just as biological tissues achieve complex shape changes via differential growth and differential muscular contraction, one may achieve a complex shape change by programming a differential patterned shape change, for example directions of contraction into an LCE [2,6–8], or magnitudes of swelling into a gel [1,9–12]. Here, we focus on initially flat sheets encoded with a pattern that varies in plane, leading to a new metric on actuation, and morphing the flat sheet into a curved surface. There is now a growing literature on such *metric-mechanics* [13], largely focused on how to design the right pattern of actuation to achieve a desired target surface [14–22]. In this paper, we discuss how such patterns can be stitched together piecewise in a single sheet without inducing any geometric incompatibility that would lead to substantial internal stress, wrinkles and perhaps even tears. This strategy allows individual patterns to be used as building blocks for more complex shape changes, and also enables the design of multi-material sheets with interfaces between different types of active material, or even between active and passive materials. It also transpires that the interfaces themselves are interesting, as they generically bear singular intrinsic curvature and form ridges in the final surface.

As a motivating example, consider an LCE sheet with a planar nematic director **n**, that encodes the direction of molecular alignment. Upon heating, alignment is disrupted by the nematic/isotropic phase transition, and the sheet contracts by $\lambda_\parallel \sim 0.7$ parallel to **n**, and expands laterally by $\lambda_\perp \sim 1/\sqrt{\lambda_\parallel} \sim 1.2$, as seen in figure 1$a$. In LCEs, programming is achieved via a spatial pattern of alignment **n(x)**, while the magnitudes of $\lambda_\parallel$ and $\lambda_\perp$ are homogeneous. For example, as seen in figure 1$b$, if a sheet is prepared with the alignment in concentric circles, heating will produce a cone [6,23]. Importantly, Gauss's *theorema egregium* tells us that the Gaussian (intrinsic) curvature of a surface cannot be altered without changing the metric [24,25]. Here, actuation does change the metric, and, accordingly, there is a point of singular Gaussian curvature (GC) at the tip of the cone. It also follows that the cone's actuation may not be blocked without an energetically prohibitive stretch, and indeed LCE cones are powerful lifters that can lift heavy loads thousands of their own weight as they rise [2,26,27]. Having designed the concentric-circle pattern, one may then seek to combine two concentric-circle patterns in a single sheet, to make a sheet that actuates to a shape with two tips. The patterns may be combined by stitching them together piecewise along a seam, but, to avoid large internal stresses, such an interface cannot be chosen arbitrarily, as both patterns must agree on the length of the interface after actuation. In LCEs, this condition is satisfied if the directors on either side make equal angles with the interface [28], leading to a pattern like figure 1$c$, which indeed makes a pair of cones on actuation [29,30]. Stitching patterns is thus a delicate constrained problem, but of vital interest as it can dramatically increase the design space.

An active sheet's metric is a $2 \times 2$ symmetric matrix, and hence has three degrees of freedom. Thus our notation using $\lambda_\parallel$, $\lambda_\perp$ and **n** can actually capture an arbitrary metric if all three quantities are able to vary spatially, so it can describe any type of active sheet. However, in most engineered soft-matter systems, one cannot pattern all three components, but rather has a restricted palette based on the material in question. In LCEs, typically **n** is patterned and $\lambda_\parallel$, $\lambda_\perp$ are homogeneous, while, at the other end of the spectrum, in isotropic gels, the actuation factors are equal ($\lambda_\parallel = \lambda_\perp$)

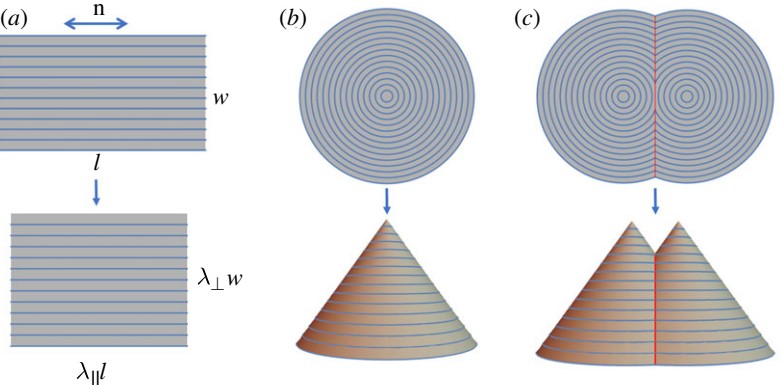

**Figure 1.** A nematic solid sheet contracts along a programmed alignment (blue lines) upon actuation. (*a*) A constant alignment n morphs a flat sheet to a flat sheet. (*b*) A concentric-circle alignment morphs a disc to a cone. (*c*) Two concentric-circle patterns with an interface (red) morph to twinned cones stitched together with a curved crease (red). (Online version in colour.)

but this single magnitude can be patterned via the crosslink density giving a conformal metric: both examples thus have a single degree of freedom for patterning. LCEs can also actuate via swelling, giving two homogeneous actuation strains that are both elongations [31], and pneumatic analogues of LCEs called baromorphs have also been designed, which actuate with $\lambda_\parallel = 1$ and $\lambda_\perp = 2/\pi$ on inflation [32]. Recent work has highlighted active systems with more than one local patterning variable. For example, an LCE with patterned **n** can then undergo actuation with patterned magnitude via a patterned temperature field [33], and, perhaps in the future, a patterned crosslink density or composition, while gel nets can be created with essentially arbitrary metric changes on swelling [34]. In all such cases, one may seek to stitch patterns together. Moreover, one inevitably faces such stitching problems whenever one has an interface between different active materials, or between active and passive material: themes which are likely essential for future work on multi-functional sheets, and reconfigurable sheets.

In the first section of the paper, we thus systematically study how to stitch together patterns of shape change in active sheets. The general condition of geometric compatibility is inherited from the well-known rank-1 compatibility condition [35–37] used in studies of twinning and microstructure in crystaline systems, including in LCEs [38–40]. Beyond describing the physics of interfaces, in LCEs this metric compatibility condition has also frequently been used for pattern design between regions of homogeneous director, where it gives rise to the basic rules of *non-isometric origami* [28,29,41,42], but the consequences for combining patterns which themselves have varying actuation are much less well studied [30]. Here, we formulate and solve the generalized two-dimensional metric-compatibility condition, for an interface between any two types of active sheet. Given a pair of patterns to join, we find that in systems of contraction/elongation like LCE sheets there is generically an infinite set of compatible interfaces available in which the director and hence the metric is discontinuous across the interface, and may also be a small finite number of possible interfaces in which the metric is continuous across the interface. By contrast, in systems of patterned isotropic swelling/growth/dilation, one has only the small finite number of the latter type, along which the swelling factors are equal in both patterns, greatly limiting the potential of stitching as a strategy. In the second part of the paper, we thus focus on LCE actuation and find, as an illustration, all the possible interfaces between pairs of logarithmic spiral director patterns (constant-speed +1 defects) which would actuate individually to anti/cones. We also use simulations to visualize the resultant actuated shapes.

As seen in the double-cone pattern in figure 1*c*, stitched metric-compatible interfaces generically also encode a singular GC, leading (ideally) to a sharp curved fold in the target surface, where one of the principal curvatures diverges. As mentioned earlier, an LCE cone

can serve as a strong lifter, as the cone tip bears singular GC, preventing the cone from being flattened. Likewise, here, the actuated interfaces cannot be flattened, in strong contrast to the superficially similar creases with zero GC seen in curved-fold origami [43–48]. As a starting point towards understanding the geometric and mechanical properties of these interfaces, we derive analytical formulae for the GC concentrated along them. These analytic results highlight that the interfaces can form folds with positive or negative GC [49] (or even both), and give a first-order understanding of the surface's resultant shape, which we further illustrate with simulations.

## 2. Metric-compatible interfaces between patterns of active shape change in sheets

### (a) Metric compatibility between two homogeneous deformations

We first consider a flat sheet that undergoes a spontaneous deformation as seen in figure 1a,

$$\mathbf{U} = \lambda_\parallel \mathbf{n} \otimes \mathbf{n} + \lambda_\perp \mathbf{n}_\perp \otimes \mathbf{n}_\perp, \tag{2.1}$$

where $\mathbf{n}$ is a unit vector in the plane, along which the material will stretch by a factor $\lambda_\parallel$, and $\mathbf{n}_\perp = \mathbf{R}(\pi/2)\mathbf{n}$ is perpendicular to $\mathbf{n}$ in the plane of the sheet ($\mathbf{R}(.)$ being a two-dimensional anticlockwise rotation) along which the material will stretch by a factor $\lambda_\perp$. Before actuation, an infinitesimal vector in the plane d$l$ has length d$l$ given by d$l^2 = $ d$l \cdot \mathbf{I}$d$l$, meaning that the initial metric is simply the identity, $\mathbf{I}$. After actuation, the same vector has length d$l_A = |\mathbf{U}$d$l|$, or, in metric form

$$\mathrm{d}l_A^2 = \mathrm{d}l \cdot \mathbf{U}^\mathrm{T}\mathbf{U}\,\mathrm{d}l = \mathrm{d}l \cdot (\lambda_\parallel^2 \mathbf{n} \otimes \mathbf{n} + \lambda_\perp^2 \mathbf{n}_\perp \otimes \mathbf{n}_\perp)\mathrm{d}l, \tag{2.2}$$

from which we identify the activated metric as $\mathbf{a} = \mathbf{U}^\mathrm{T}\mathbf{U} = \lambda_\parallel^2 \mathbf{n} \otimes \mathbf{n} + \lambda_\perp^2 \mathbf{n}_\perp \otimes \mathbf{n}_\perp$.

We now consider the unactuated sheet contains two regions with spontaneous deformations $\mathbf{U}_i$ ($i = 1, 2$) with different actuation parameters $\mathbf{n}_i$, $\lambda_\parallel^{(i)}$, $\lambda_\perp^{(i)}$, as shown in figure 2a. If the interface between these regions has unit tangent $\mathbf{t}$ in the unactuated state, then in the actuated configuration, the lengths of the unit tangent actuated from the two fields are $|\mathbf{U}_1\mathbf{t}|$ and $|\mathbf{U}_2\mathbf{t}|$, respectively. For the interface to be compatible, these actuated interfaces need to have equal length. This metric compatibility condition reads

$$|\mathbf{U}_1\mathbf{t}| = |\mathbf{U}_2\mathbf{t}| \Leftrightarrow \mathbf{t} \cdot \mathbf{a}_1\mathbf{t} = \mathbf{t} \cdot \mathbf{a}_2\mathbf{t}, \tag{2.3}$$

where $\mathbf{a}_i = \mathbf{U}_i^\mathrm{T}\mathbf{U}_i$ as defined above. Following figure 2a, we denote the angle between $\mathbf{n}_1$ and $\mathbf{n}_2$ as $\xi$, and the angle between $\mathbf{t}$ and $\mathbf{n}_1$ by $\theta$. Inserting these into the condition above yields a quadratic equation for $\tan\theta$ which we must solve to find metric-compatible directions

$$a\tan^2\theta + b\tan\theta + c = 0,$$

where

$$a = (\lambda_\perp^{(1)})^2 - (\lambda_\perp^{(2)})^2 - \left[(\lambda_\parallel^{(2)})^2 - (\lambda_\perp^{(2)})^2\right]\sin^2\xi,$$

$$b = -2\left[(\lambda_\parallel^{(2)})^2 - (\lambda_\perp^{(2)})^2\right]\sin\xi\cos\xi \tag{2.4}$$

and

$$c = (\lambda_\parallel^{(1)})^2 - (\lambda_\perp^{(2)})^2 - \left[(\lambda_\parallel^{(2)})^2 - (\lambda_\perp^{(2)})^2\right]\cos^2\xi.$$

Given values for the actuation parameters, one may simply solve this quadratic equation to find either 0, 1, 2 or, in degenerate cases, infinitely many solutions for $\tan\theta$, which describe the metric-compatible directions. Compatible interfaces between homogeneous deformations are thus straight lines. One may characterize the equation algebraically, but a graphical approach is highly instructive for understanding these different cases and constructing such stitched interfaces between patterns. In figure 2b, the peanut-shaped blue curve shows, for each reference-state direction $\hat{\mathbf{d}}$, the length $|\mathbf{U}\hat{\mathbf{d}}|$ attained after actuation. The curve is aligned with $\mathbf{n}$ and has radius $\lambda_\parallel$ along $\mathbf{n}$ and $\lambda_\perp$ orthogonally. At the interface, we have two spontaneous deformations, giving rise to two peanut-shaped curves, each with different stretches and oriented along its

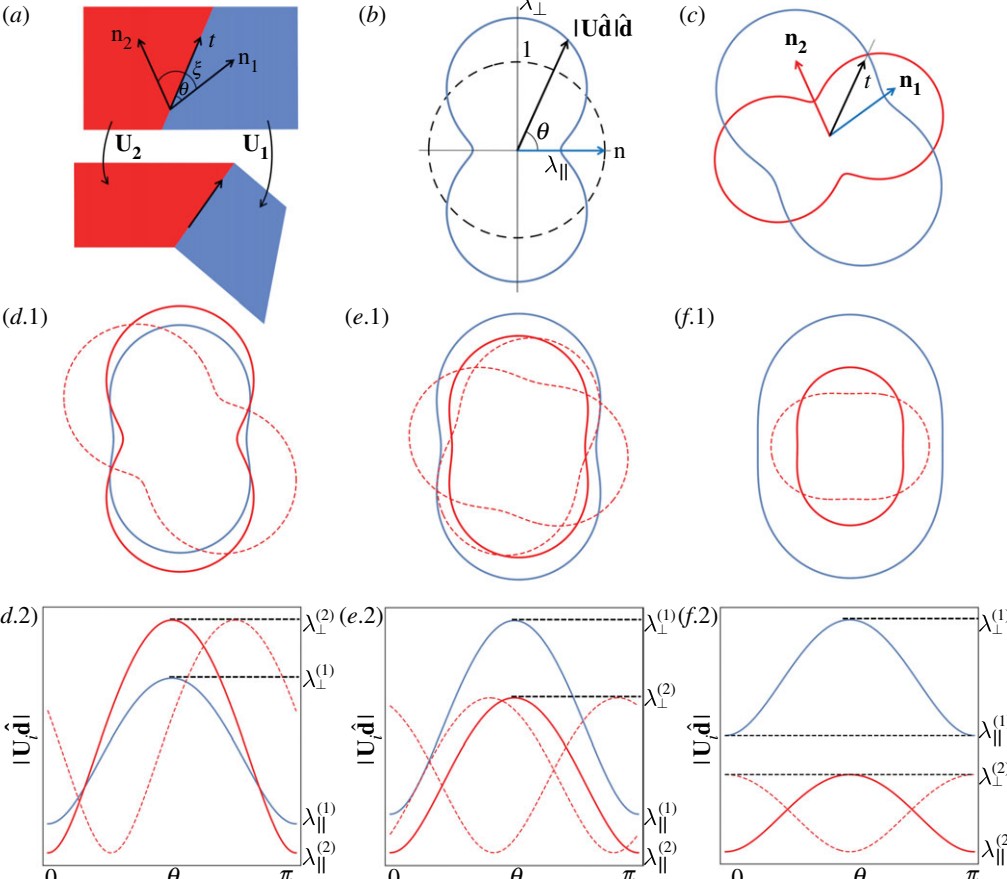

**Figure 2.** (*a*) Metric compatibility between two homogeneous spontaneous deformations $\mathbf{U}_1$ (blue) and $\mathbf{U}_2$ (red). (*b*) The actuated length of each reference-state direction $\hat{\mathbf{d}}$, plotted in the reference state by mapping $\hat{\mathbf{d}}$ to $|\mathbf{U}\hat{\mathbf{d}}|\hat{\mathbf{d}}$. (*c*) Two homogeneous deformations $\mathbf{U}_1$ and $\mathbf{U}_2$ generate two peanut-shaped curves; the intersections correspond to metric-compatible interfaces. (*d–f*) Three scenarios for metric-compatible interfaces, in which the range of spontaneous stretches of $\mathbf{U}_2$ encompasses (*d*), overlaps with (*e*), and does not overlap with (*f*) that of $\mathbf{U}_1$. In each scenario, we plot the peanut curves for specific values of the spontaneous stretches, but a range of angles $\xi$ between the $\mathbf{n}_1$ and $\mathbf{n}_2$, to explore when intersections arise. Below, we also plot the same functions as linear rather than polar plots (*d*.2–*f*.2), in which case rotation becomes translation. In (*d*), there are always four intersections and two compatible interfaces, in (*e*), there can be two or zero depending on $\xi$, and in (*f*) there are always zero. (Online version in colour.)

own $\mathbf{n}$, as seen in figure 2*c*. Metric-compatible directions are given by the intersections, where both metrics agree on actuated length; there are four such intersections in figure 2*c*, indicating two compatible directions since $\mathbf{t}$ and $-\mathbf{t}$ describe the same interface. To explore the different possibilities, we suppose, without loss of generality, that the actuation strains are ordered in each material such that $\lambda_\parallel \leq \lambda_\perp$ (as is natural for LCEs on heating), that region two has the lower parallel actuation factor $\lambda_\parallel^{(2)} \leq \lambda_\parallel^{(1)}$. We may then distinguish four cases:

(i) If $\lambda_\perp^{(2)} \geq \lambda_\perp^{(1)}$ (figure 2*d*), the range of strains in region two fully encloses the range of strains in region one. There are always four intersections, for any $\xi$, and hence two metric-compatible interfaces. This is easily seen by considering the plot figure 2*d*.2, in which changing $\xi$ simply translates the red curve.

(ii) If $\lambda_\parallel^{(1)} \leq \lambda_\perp^{(2)} < \lambda_\perp^{(1)}$ (figure 2*e*), the strain ranges are partially overlapping. In this case, there can be two, one or zero interface, depending on the angle $\xi$ between $\mathbf{n}_1$ and $\mathbf{n}_2$. The critical $\xi$ corresponds to one interface and can be determined by solving $b^2 - 4ac = 0$ in (2.4).

(iii) If $\lambda_\perp^{(2)} < \lambda_\parallel^{(1)}$ (figure 2*f*), the ranges of strains in each metric do not overlap, so there is no metric-compatible interface for any $\xi$.

(iv) If $\mathbf{U}_1 = \mathbf{U}_2$, the two peanut-shaped curves are identical so every direction is compatible.

## (b) Compatible interfaces between patterns

The above considerations allow one to find the metric-compatible directions between a pair of specific spontaneous deformations, $\mathbf{U}_1$ and $\mathbf{U}_2$. We now consider stitching together two patterns of actuation $\mathbf{U}_1(\mathbf{x})$ and $\mathbf{U}_2(\mathbf{x})$. Our approach is to overlay the patterns in the reference domain and solve equation (2.4) at each point to find the set of compatible directions $\{\mathbf{t}_i(\mathbf{x})\}$ at each reference point $\mathbf{x}$, where $i$ enumerates the set of solutions at each point. We may then construct a compatible arc-length parameterized interface $\mathbf{x}(l)$ between the patterns by starting at any point where a solution exists, choosing a $\mathbf{t}_i(\mathbf{x})$ then propagating the solution in this compatible direction by solving the ODE

$$\mathbf{x}'(l) = \mathbf{t}_i(\mathbf{x}(l)). \tag{2.5}$$

In general, this approach will yield a curved interface. The solution to the ODE may terminate by reaching the boundary, forming a loop, or reaching a region where $\mathbf{t}_i(\mathbf{x})$ no longer exists.

We note that while the condition of metric compatibility is a necessary condition for a stitched metric to be embeddable in three dimensions, it is not clear whether it is a sufficient condition. It is possible that a stitched sheet still cannot find an actuated configuration that follows the design metric, and will instead have to undergo some stretch on actuation. However, it is also unclear what more could be demanded of a metric-compatible interface during stitching. Furthermore, as will be seen throughout this paper, it is our experience in simulations that stitched sheets actuate to surfaces that are very close to the design metric, suggesting that well-behaved embeddings are generically available. We now consider three common scenarios in active soft sheets.

### (i) Interface between two gel patterns

First, we consider interfaces between two actuation patterns of swelling-gel type, with different but isotropic actuation strains $\lambda_\parallel^{(i)} = \lambda_\perp^{(i)} \equiv \lambda^{(i)}$. At a generic material point, the actuation strains will be different in the two patterns, so we are in case 3, and no directions are compatible.

However, if the patterns are continuous in the reference domain, there are likely to be a finite number of lines in the reference domain where the actuation strains are equal, $\lambda^{(1)} = \lambda^{(2)}$. To see this, consider that if one finds a reference point with $\lambda^{(1)} > \lambda^{(2)}$ and another with $\lambda^{(1)} < \lambda^{(2)}$, the intermediate value theorem implies there is a point of equality on any reference path between the two points, so the set of such points will indeed form lines. At any point along such a line, we are in case 4, and all directions are compatible. However, we may only integrate equation (2.5) along the line itself, as this is where the solutions exist, giving rise to a finite number of *continuous-metric interfaces*.

### (ii) Interface between two liquid crystal elastomers patterns

Second, we consider interfaces between two patterns of LCE actuation with homogeneous and equal actuation strains $(\lambda_\parallel, \lambda_\perp)$ but different directors $\mathbf{n}_1(\mathbf{x})$ and $\mathbf{n}_2(\mathbf{x})$. In this case, the compatibility condition reduces to the simpler condition

$$|\mathbf{n}_1 \cdot \mathbf{t}| = |\mathbf{n}_2 \cdot \mathbf{t}|. \tag{2.6}$$

At a generic reference point $\mathbf{x}$ we will be in case 1, and there are two metric-compatible directions, which in this case are the bisector of $\mathbf{n}_1$ and $\mathbf{n}_2$ (i.e. $\theta = \xi/2$) and the orthogonal direction

($\theta = \xi/2 + \pi/2$), giving the tangent fields

$$\mathbf{t}_1 = \pm \frac{(\mathbf{n}_1 + \mathbf{n}_2)}{|\mathbf{n}_1 + \mathbf{n}_2|} \quad \text{and} \quad \mathbf{t}_2 = \pm \frac{(\mathbf{n}_1 - \mathbf{n}_2)}{|\mathbf{n}_1 - \mathbf{n}_2|}. \tag{2.7}$$

To find an interface, we simply start at any point, choose one of the two available tangent directions, and integrate equation (2.5) to find the interface curve. The curves are thus the integral curves of $\mathbf{t}_1$ and $\mathbf{t}_2$, and form a full orthogonal coordinate system over the pattern domain (just like the director and its dual [50]), with two orthogonal interfaces passing through each material point. We call interfaces constructed along these directions *twinning interfaces*, as the directors are discontinuous at the interface but symmetrically satisfy the twinning condition $|\mathbf{n}_1 \cdot \mathbf{t}| = |\mathbf{n}_2 \cdot \mathbf{t}|$. Then the deformations $\mathbf{U}_{1,2}(\mathbf{x})$ are mirrors of each other across the interface. An example is seen in figure 1c.

There may also be reference points where the two directors are equivalent $\mathbf{n}_1 = \pm\mathbf{n}_2$. At these points, we are in case 4, and all directions are compatible. However, the intermediate value theorem (this time applied to the difference in director angle) again shows that such points generically occur as isolated lines in the reference state, and we may only integrate along the line where solutions exist. This process yields a finite number of extra *continuous-metric interfaces*, across which the patterns can be joined without any discontinuity in the resultant director.

### (iii) Interface between active and passive material

Thirdly, we consider interfaces between a generic active material $\mathbf{U}_1$ and a passive material with $\mathbf{U}_2 = \mathbf{I}$. At a given material point, if the active actuation strains span unity, $\lambda_\parallel < 1 < \lambda_\perp$, then we will be in case 1, with two compatible directions. The compatible direction then lies along a critical angle $\theta$ [51] which is the angle between compression and extension in the active material where length is unchanged

$$\tan(\theta) = \sqrt{\frac{1 - \lambda_\parallel^2}{\lambda_\perp^2 - 1}}. \tag{2.8}$$

This situation is generic for LCE/passive boundaries. As with twinned boundaries in LCEs, the angle $\theta$ also has two solutions and these two directions will again generate a coordinate-system-like set of integral curves, with two curves through each material point, although in this case, the two sets of curves are not orthogonal. However, clearly, such interfaces are impossible for gels, where the actuation strains are equal, except at material points where $\mathbf{U}_1 = \mathbf{I}$, i.e. where the active material is not actually strained at all. As with other *continuous-metric interfaces*, these will generically occur along a finite number of lines in the pattern.

Curiously, none of these standard systems explores case 2, although this situation would arise naturally at the interface between two LCEs with different actuation strain magnitudes.

## 3. Examples of metric-compatible interfaces between two liquid crystal elastomers patterns

A key conclusion from these considerations is that stitching is a powerful tool in LCE systems, where there are typically infinitely many twinned interfaces available, but is a very limited tool in isotropic systems where one typically has only a finite set of possible interfaces, and possibly even none at all. For the remainder of the paper, we thus focus on LCE systems, and demonstrate the method by explicitly finding all the compatible interfaces between some pairs of well-studied patterns of LCE actuation. Throughout, we complement our stitched patterns with numerical simulations of the resultant surfaces, computed by minimizing a full shell energy (stretch plus bend) following the approach in [52] and the electronic supplementary material, S2.

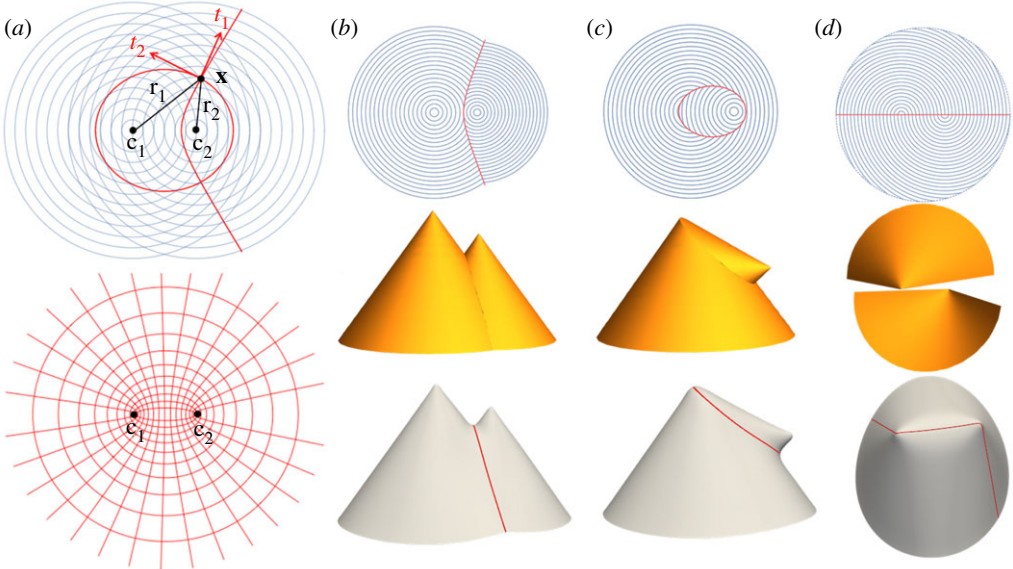

**Figure 3.** (*a*) Top: two overlaid concentric-circle patterns. At a generic point **x**, there are two compatible tangent directions $\mathbf{t}_1$, $\mathbf{t}_2$. The integral curves of these tangent vectors form two compatible twinned interfaces: a hyperbola and an ellipse respectively. Bottom: the set of all such curves forms the standard elliptic/hyperbolic coordinate system. (*b,c*) Examples of hyperbolic (*b*) and elliptic (*c*) interfaces. The yellow surfaces are exact isometries constructed from the individual conical deformations, and closely match the simulations shown below. (*d*) There is also one continuous-metric interface along the horizontal. In this case, the two individual conical deformations do not fit together, but our simulation shows that the sheet can nonetheless reach an isometry with two tips. (Online version in colour.)

## (a) Interfaces between concentric-circle patterns

As a simple first example, we take a pair of concentric-circle patterns, with the centres at position vectors in $(x, y)$ Cartesian coordinates given by $\mathbf{c}_1 = (c, 0)$ and $\mathbf{c}_2 = (-c, 0)$, as shown in figure 3*a*. A generic point **x** has vector separation from each centre $\mathbf{r}_1 = \mathbf{x} - \mathbf{c}_1$ and $\mathbf{r}_2 = \mathbf{x} - \mathbf{c}_2$, respectively, which point in the radial direction of each circle pattern, orthogonal to the director. The two twinned-interface tangent directions at **x**, $\mathbf{t}_1$ and $\mathbf{t}_2$ are thus the bisector of $\mathbf{r}_1$ and $\mathbf{r}_2$, and its orthogonal dual. Using the focus-to-focus reflection property of ellipses, one can show that the compatible twinned interfaces are the set of ellipses and hyperbolae with foci at the pattern centres [30]. As expected, there are two such interfaces through each point, and the set of curves forms an orthogonal coordinate system—in this case, the standard elliptic/hyperbolic coordinate system, as shown in figure 3*a* bottom. The straight interface in figure 1*c* is a special case of a hyperbolic interface. More generic examples of hyperbolic and elliptic interfaces are shown in figure 3*b,c*.

An individual concentric-circle pattern actuates to a cone. Furthermore, one may show that the interfaces are not just metrically compatible, but also 'conically consistent', meaning they would actuate to the same shape under each individual conical deformation [30]. This means one may construct analytic isometries of the stitched metric by combining conical surfaces: the hyperbolic interface leads to a pair of cones with different heights, while the elliptic interface yields a cone with its tip replaced by one from the other pattern.

In addition to these twinned interfaces, there is also a single continuous-metric interface available along the horizontal, as seen in figure 3*d*. In this case, the interface is not conically consistent so we cannot construct an analytic shape, but simulation reveals a shape with a tip at each centre.

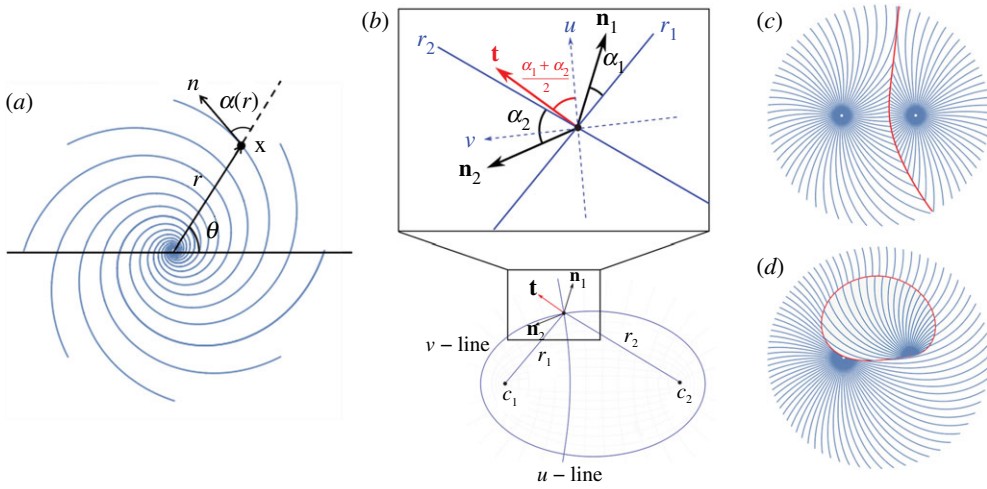

**Figure 4.** (*a*) A director pattern with rotational symmetry, described by the angle $\alpha(r)$ between the nematic director **n** and the radial direction. (*b*) Illustration of the twinning interface between two directors $\mathbf{n}_1$ and $\mathbf{n}_2$. The compatible direction **t** bisects $\mathbf{n}_1$ and $\mathbf{n}_2$, and makes an angle $(\alpha_1 + \alpha_2)/2$ with the local hyperbola whose tangent is **u**. (*c*) A twinning interface between two rotationally invariant director patterns with $\alpha_1(r) = r^{3/2}$ and $\alpha_2(r) = -r^2$. (*d*) A continuous-metric interface between two rotationally invariant director patterns with $\alpha_1(r) = r^2$ and $\alpha_2(r) = -r$. (Online version in colour.)

## (b) Interfaces between patterns with rotational symmetry

In LCE programming, there has also been substantial work on patterns with rotational symmetry [18,19,23,32,53,54]. Working in plane-polar coordinates $(r, \theta)$, these patterns are conventionally described by the angle $\alpha(r)$ between the nematic director and the radial direction, as shown in figure 4*a*. Explicit forms of $\alpha(r)$ have been given for (anti)cones, (pseudo)spherical caps and spindles, and even a generic surface of revolution [19]. Here, we find interfaces between two such patterns, again centred at $\mathbf{c}_1$ and $\mathbf{c}_2$, and described by $\alpha_1(r_1)$ and $\alpha_2(r_2)$, where $r_i = |\mathbf{r}_i|$, reusing our earlier notation.

Inspired by the circle case, we describe the reference plane using elliptic/hyperbolic coordinates focused on the pattern centres, by setting $(x, y) = c(\cosh u \cos v, \sinh u \sin v)$, such that $\mathbf{x}(u_0, v)$ and $\mathbf{x}(u, v_0)$ define a $v$-line (ellipse) and a $u$-line (hyperbola), respectively, as shown in figure 4*b*, bottom. As in the circle case, at a generic point $\mathbf{x}$, the local hyperbola bisects $\mathbf{r}_1$ and $\mathbf{r}_2$. However, now the local directors make angles $\alpha_1(r_1)$ and $\alpha_2(r_2)$ with these radial directions, so the two compatible directions, (the bisector **t** of the directors, and its orthogonal dual) make angles $(\alpha_1 + \alpha_2)/2$ and $(\alpha_1 + \alpha_2)/2 + \pi/2$ with the local hyperbola (figure 4*b*, top). Integrating to propagate the interface leads to a differential equation for each compatible direction:

$$\mathbf{t}_1 : \quad \frac{\mathrm{d}v}{\mathrm{d}u} = \tan\left(\frac{\alpha_1(r_1) + \alpha_2(r_2)}{2}\right) \quad \text{and} \quad \mathbf{t}_2 : \quad \frac{\mathrm{d}v}{\mathrm{d}u} = \cot\left(\frac{\alpha_1(r_1) + \alpha_2(r_2)}{2}\right). \tag{3.1}$$

The distances $r_1$ and $r_2$ in elliptic coordinates can be conveniently expressed as $r_1 = c(\cos v + \cosh u)$ and $r_2 = c(-\cos v + \cosh u)$. Therefore, both equations (3.1) are first-order ODEs in $(u, v)$, which can be solved efficiently using numerical methods, for any form of $\alpha_1$ and $\alpha_2$. The circle case is recovered by setting $\alpha_1 = \alpha_2 = \pi/2$, so the equations trace out $u$ and $v$ lines, respectively. A more sophisticated example is given in figure 4*c*, showing that the directors are indeed twinned at the interface between two rotationally invariant patterns.

There may also be a finite number of continuous-metric interfaces. Scrutinizing figure 4*a*, we see that the director will be continuous if satisfying $\alpha_1(r_1) + \theta_1 = \alpha_2(r_2) + \theta_2 + k\pi$, $k \in \mathbb{Z}$. Taking the tan of both sides and then substituting for $\theta_i$ using elliptic/hyperbolic coordinate system, we

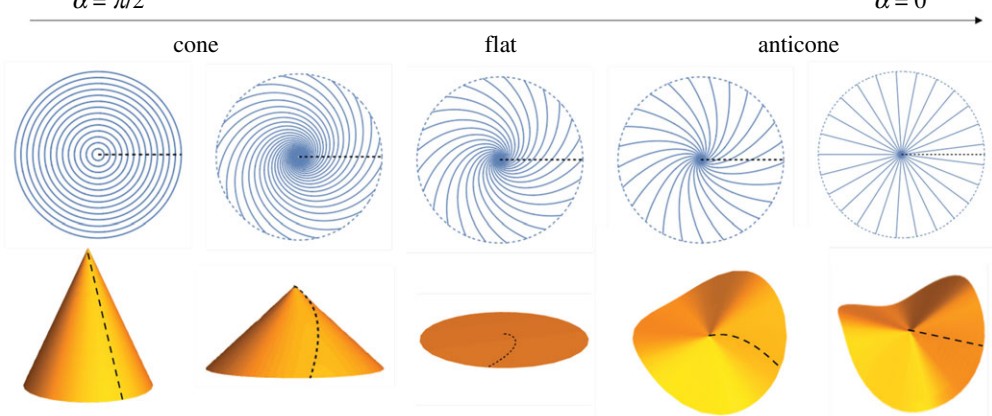

$\alpha = \pi/2$ $\alpha = 0$

cone flat anticone

**Figure 5.** The log-spiral patterns actuate to (anti)cones, varying from cone to flat to anticone as $\alpha$ varies from $\alpha = \pi/2$ to $\alpha = 0$. The reference and actuated black dashed lines indicate that the actuations involve twisting when $\alpha \neq 0, \pi/2$. (Online version in colour.)

obtain

$$(\sin^2 v - \sinh^2 u)\sin[\alpha_1(r_1) - \alpha_2(r_2)] + 2\sin v \sinh u \cos[\alpha_1(r_1) - \alpha_2(r_2)] = 0. \tag{3.2}$$

An example is given in figure 4d, showing that the directors are continuous across the interface. However, the existence of such a curve is not guaranteed, and must be addressed case-by-case.

### (c) Interfaces between logarithmic spiral patterns

A particularly well-studied set of patterns with rotational symmetry are those with constant $\alpha$, yielding surfaces that are Gauss-flat except for a singular point of the origin. For $\alpha = \pi/2$, these patterns are circles and actuate to make cones, while for $\alpha = 0$ the patterns are radii, leading to ruff-like anticones with a point of negative GC. For intermediate $\alpha$, these patterns are log-spirals, whose actuated shapes vary from cone to flat to anticone as $\alpha$ varies from $\alpha = \pi/2$ to $\alpha = 0$. However, in contrast to the circular patterns, the conical deformations for the log-spiral patterns (given explicitly in the electronic supplementary material, S3) involve twisting about the cone axis, as illustrated in figure 5.

For a pair of log-spirals, we may easily integrate the ODEs in equation (3.1) analytically to find the form of the twinned interfaces. The first, $\mathbf{t}_1$, which bisects $\mathbf{n}_1$ and $\mathbf{n}_2$, is simply $v(u) = u\tan((\alpha_1 + \alpha_2)/2) + v_0$ in elliptic coordinates, and correspondingly, in Cartesian coordinates has the form

$$\left.\begin{aligned} x(u) &= c\cosh u \cos\left(u\tan\frac{\alpha_1 + \alpha_2}{2} + v_0\right) \\ \text{and} \qquad y(u) &= c\sinh u \sin\left(u\tan\frac{\alpha_1 + \alpha_2}{2} + v_0\right) \end{aligned}\right\}. \tag{3.3}$$

The orthogonal twinning interface, which bisects $-\mathbf{n}_1$ and $\mathbf{n}_2$, can be solved similarly, and reads

$$\left.\begin{aligned} x(v) &= c\cosh\left(-v\tan\frac{\alpha_1 + \alpha_2}{2} + u_0\right)\cos v \\ \text{and} \qquad y(v) &= c\sinh\left(-v\tan\frac{\alpha_1 + \alpha_2}{2} + u_0\right)\sin v \end{aligned}\right\}. \tag{3.4}$$

In the above equations, $u_0$ and $v_0$ are constants that ensure the interface starts at the desired point. As expected, these two sets of twinned interfaces are orthogonal, and one of each goes through each reference point. Two examples between a spiral pattern with $\alpha_1 = 5\pi/12$ and a

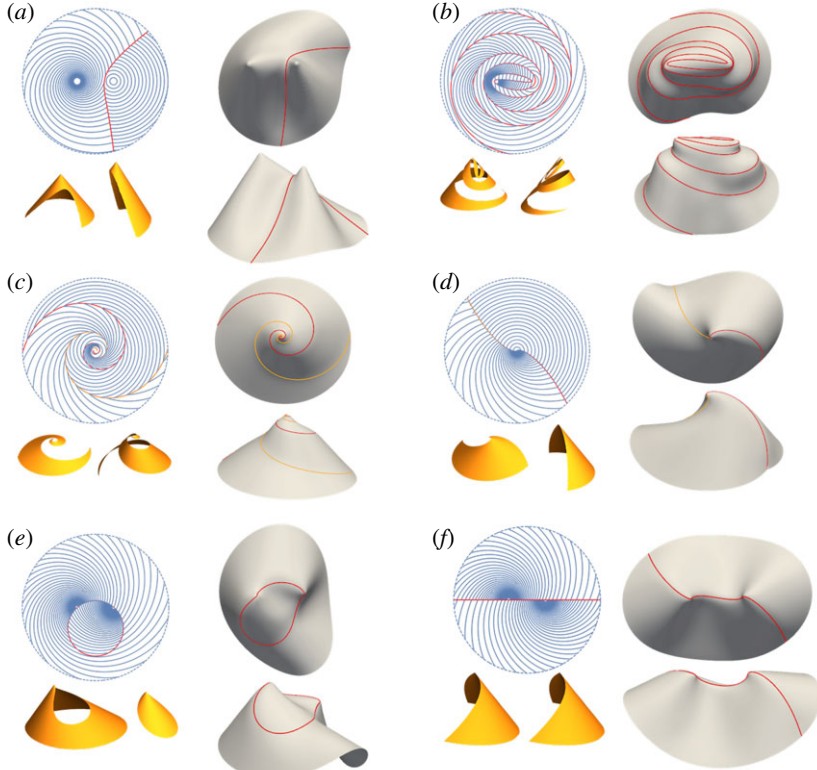

**Figure 6.** Examples of metric-compatible interfaces between constant-$\alpha$ axisymmetric patterns. (*a–f*) Top left: reference domains. Bottom left: analytical individual conical parts. Top right and bottom right: two views of the simulated configurations. (*a,b*) Generic twinning interfaces for $\alpha_1 = 5\pi/2$ and $\alpha_2 = \pi/2$. (*c,d*) Twinning interfaces for concentric spiral patterns ($\alpha_1 = 1$ and $\alpha_2 = \pi/2$) which degenerate to concentric spirals. (*e,f*) The continuous-metric interface between two spiral patterns is a circle when the spirals are different ((*e*), $\alpha_1 = 1.173$, $\alpha_2 = -1.249$), or the horizontal axis when the spirals are identical ((*f*), $\alpha_1 = \alpha_2 = 1.173$). (Online version in colour.)

circular pattern with $\alpha_2 = \pi/2$ are shown in figure 6*a,b*. In these cases, we see that neither interface forms a closed loop, rather they spiral out to infinity. In figures 6*c,d*, we also highlight an interesting special case, in which the pattern centres coincide, and the twinned interfaces themselves degenerate to log-spirals from the single centre. We provide more special cases in figure S1 of the electronic supplementary material for certain choices of $\alpha_1$ and $\alpha_2$.

Continuous-metric interfaces also exist, obeying the analytical form (3.2). By substituting constant $\alpha_1$ and $\alpha_2$, we find that the interface is a circle of radius $c/|\sin(\alpha_1 - \alpha_2)|$ passing through the centres when the log-spirals are different ($\alpha_1 \neq \alpha_2$), which degenerates to a straight line when the log-spirals are identical ($\alpha_1 = \alpha_2$), as shown in figure 6*e,f* respectively.

In all cases, the actuations of the individual regions constructed using the single-pattern conical isometries (given in the electronic supplementary material, S3) do not give consistent shapes for the interfaces (yellow surfaces in figure 6)—they are not conically consistent—so we cannot construct isometries of the actuated sheets analytically. However, we present shapes from simulations that confirm that isometries do exist, with the actuated surfaces containing tips at the centres and a curved fold along the interface.

## 4. Gaussian curvature concentrated along creases

In our analytic forms for the actuated surfaces, we observe that the interfaces generically actuate to form sharp creases that carry singular GC, and there are corresponding ridges in our numerics.

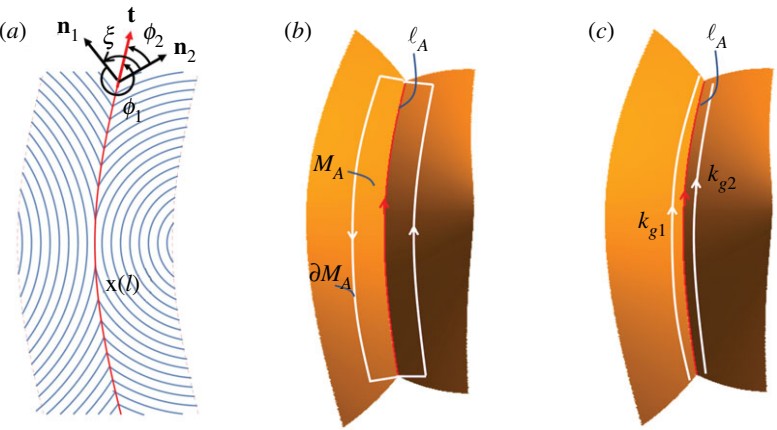

**Figure 7.** (a) The unit tangent **t** of the arc-length parameterized twinning interface **x**($l$) in the reference configuration is expressed as $\mathbf{t} = \cos\phi_i\mathbf{n}_i + \sin\phi\mathbf{n}_i^\perp$ for $i = 1, 2$, where $\phi_i$ is the angle from the director $\mathbf{n}_i$ to the interface. The twinning angle $\xi$ is the angle between $\mathbf{n}_1$ and $\mathbf{n}_2$ across **t**. (b) The Gauss–Bonnet loop composed of solid white lines is the boundary of the patch $M_A$ containing the actuated interface $l_A$. (c) To characterize the GC on the crease, we take the width of the patch to zero. In the resulting formula equation (4.2), we evaluate the geodesic curvatures $\kappa_{g_1}$ and $\kappa_{g_2}$ in the sense shown by the white arrows. (Online version in colour.)

For example, the actuated twinning interface between two circular patterns in figure 3b carries negative GC, since the curvature of the interfacial arc is in the opposite sense to that of the fold. As a result, the crease itself cannot be flattened into the plane without stretching, making it dramatically different to, and potentially much stronger than, the curved folds seen in isometric origami [55,56]. As a starting point towards understanding the geometry and mechanics of these intrinsic creases, we first use the Gauss–Bonnet theorem to derive the expression for the distribution of GC along such a crease based on its final state embedding, then show how the answer can be computed in the reference state for a metric-compatible interface between LCE patterns. Finally, we compute the distribution for two examples of stitched log-spiral patterns, and show that the analytic form is very helpful for predicting and interpreting the simulated shape.

## (a) Quantifying Gaussian curvature on a crease

Consider the LCE director pattern with an interface **x**($l$) in figure 7a and its actuated configuration in figure 7b. The actuated surface contains a crease (red) that follows a curve $l_A$ through three-dimensional space, which we may parameterize by its (activated) arc-length $l_A$. Although the GC on the crease, $K_A$, is singular, the Gauss–Bonnet theorem guarantees that the total curvature $\Omega \equiv \int K_A \, dA_A$, is finite. More precisely, the Gauss–Bonnet theorem says that the total curvature within any patch $M_A$ of the activated surface may be computed as [25]

$$\iint_{M_A} K_A \, dA_A = 2\pi \chi(M_A) - \oint_{\partial M_A} \kappa_g \, ds_A - \Sigma \beta_i, \tag{4.1}$$

where $\chi(M_A)$ is the Euler characteristic of the patch (which is 2 for any patch that is topologically a disc), $\kappa_g$ is the geodesic curvature of the patch boundary $\partial M_A$, and $\beta_i$ are the discrete turning angles of any corners on the patch boundary. The integration measure $dA_A$ is activated area, while the loop integral of geodesic curvature is with respect to activated arc length, and conducted anticlockwise, as shown in figure 7b.

The geodesic curvature of a curve on a surface is computed by projecting its three-dimensional curvature vector into the tangent plane of the surface. To interrogate the crease, we use a long narrow patch like that shown in figure 7b. Taking the limit of the width of the patch to zero,

the long sides degenerate to the crease curve itself, although they still encode different geodesic curvatures, as the same three-dimensional curvature vector is projected into different tangent planes on the two sides. We construct the ends of the loop so that the four corners are right-angles in the actuated surface, in which case the short end caps are the shortest paths connecting the corners, and hence have zero geodesic curvature. The corner contributions thus cancel the Euler characteristic, and the geodesic integral reduces to the contributions from the two long sides, which can be combined in a single line integral

$$\Omega = \iint_{M_A} K_A \, dA_A = \int_{l_A} (\kappa_{g_1} - \kappa_{g_2}) \, dl_A. \tag{4.2}$$

Here, the sign of the $\kappa_{g_1}$ has flipped from that naively expected from equation (4.1), as we now compute it from a curve traversed in the same sense as the actuated interface $l_A$ traverses the crease (figure 7c), whereas in the original Gauss–Bonnet loop, we traversed this side in the opposite sense. The standard formulae for geodesic curvature assign the sign by comparison with the tangent normal of the curve's Darboux frame, meaning the sign flips if the curve is traversed in the opposite sense. Here, this change is helpful as the original loop integral has degenerated to a single line integral along the crease, so it is now intuitive to compute both geodesic curvatures for curves traversed in this sense.

Having computed $\Omega(l_A)$ above, we may now simply quantify the concentration of GC along the crease as the total curvature per unit length, $d\Omega/dl_A = \kappa_{g_1} - \kappa_{g_2}$.

## (b) Concentrated Gaussian curvature along metric-compatible interfaces between two director patterns

In the preceding section, the Gauss–Bonnet was done in the actuated state. However, geodesic curvature and total curvature, like GC, are actually intrinsic properties, meaning they may be computed from the metric without knowing the actual form of the surface. For active sheets, this means one can express the integral in equation (4.2) entirely in the reference state. Here, we derive such a result for the case of metric-compatible interfaces between LCEs with the same actuation magnitudes, although a similar approach could be adopted in any case.

Consider first an LCE sheet with director field $\mathbf{n}(\mathbf{x})$ in the (flat) reference state. The director variation in this state is usefully described by its two-dimensional bend and splay vectors,

$$\mathbf{b} = (\nabla \times \mathbf{n})\mathbf{n}^\perp \quad \text{and} \quad \mathbf{s} = (\nabla \cdot \mathbf{n})\mathbf{n}, \tag{4.3}$$

where the curl is taken in a two-dimensional scalar sense [50]. Note that these vectors, like the metric and the nematic phase itself, are invariant under $\mathbf{n} \to -\mathbf{n}$.

An arc-length parameterized curve $\mathbf{x}(l)$ with total length $\bar{l}$ in the reference state will become a curve on the actuated surface upon heating. The geodesic curvature of $\mathbf{x}(l)$ after actuation may be computed by an application of *Liouville's formula* (see p. 296 and p. 351 of [25]) and is given by Duffy & Biggins [52]:

$$\kappa_g = \frac{\lambda_\| \lambda_\perp}{(\mathbf{t} \cdot \mathbf{at})^{3/2}} \frac{d\phi}{dl} + \frac{(\lambda_\|/\lambda_\perp)\mathbf{b} \cdot \mathbf{t}^\perp - (\lambda_\perp/\lambda_\|)\mathbf{s} \cdot \mathbf{t}^\perp}{\sqrt{\mathbf{t} \cdot \mathbf{at}}}, \tag{4.4}$$

where $\mathbf{a}$ is the metric tensor, $\phi$ is the angle between the curve tangent $\mathbf{t}$ and the director $\mathbf{n}$ (obtained by $\mathbf{t} = \cos\phi\mathbf{n} + \sin\phi\mathbf{n}^\perp$) and $\mathbf{t}^\perp = \mathbf{R}(\pi/2)\mathbf{t}$ is the unit vector perpendicular to $\mathbf{t}$, as shown in figure 7a.

Given this expression for geodesic curvature, we now calculate the integrated GC along a metric-compatible interface $\mathbf{x}(l)$ between two director patterns. Substituting equation (4.4) into equation (4.2), and using $dl_A/dl = \sqrt{\mathbf{t} \cdot \mathbf{a}_1 \mathbf{t}} = \sqrt{\mathbf{t} \cdot \mathbf{a}_2 \mathbf{t}}$ to pass the line integral from the actuated domain to the reference domain, we obtain the integrated GC as

$$(\text{twinning case}) \quad \Omega = \iint_{M_A} K_A \, dA_A = \int_0^{\bar{l}} \left[ -\frac{\lambda_\| \lambda_\perp}{\mathbf{t} \cdot \mathbf{a}_1 \mathbf{t}} \xi'(l) + \left( \frac{\lambda_\|}{\lambda_\perp} \Delta b^\perp - \frac{\lambda_\perp}{\lambda_\|} \Delta s^\perp \right) \right] dl, \tag{4.5}$$

where $\xi(l)$ is the twinning angle between $\mathbf{n}_1$ and $\mathbf{n}_2$ across the interface, as shown in figure 7a, and $\Delta b^\perp$ and $\Delta s^\perp$ are the jumps in the perpendicular components of the bend and splay vectors across the interface, defined as

$$\Delta b^\perp = (\mathbf{b}_1 - \mathbf{b}_2) \cdot \mathbf{t}^\perp \quad \text{and} \quad \Delta s^\perp = (\mathbf{s}_1 - \mathbf{s}_2) \cdot \mathbf{t}^\perp. \tag{4.6}$$

In equation (4.5), we have suppressed the $l$ dependence in $\mathbf{a}_1, \mathbf{t}, \Delta b^\perp$ and $\Delta s^\perp$ to simplify our notation. For the continuous-metric interface along which $\mathbf{n}_1 = \pm\mathbf{n}_2$ (such as the examples in figures 6e,f), the first term in (4.4) cancels between the two sides as $d\phi_1/dl = d\phi_2/dl$, leading to the simpler result

$$\text{(continuous-metric case)} \quad \Omega = \iint_{M_A} K_A \, dA_A = \int_0^{\bar{l}} \left( \frac{\lambda_\parallel}{\lambda_\perp} \Delta b^\perp - \frac{\lambda_\perp}{\lambda_\parallel} \Delta s^\perp \right) \, dl. \tag{4.7}$$

Interestingly, we see that both types of interfaces generically bear concentrated GC. Furthermore, the different dependencies on $\lambda_\parallel$ and $\lambda_\perp$ in the three terms in equation (4.5) suggest that obtaining a perfect cancellation for a Gauss-flat interface is hard to achieve, and, even if it is achieved, is likely to hold only when actuation is complete, rather than at the intermediate actuation values en-route. Thus stitching as a design strategy is inextricably linked with intrinsically curved folds. We hope that in the future these results may allow the design of patterns that bear desired GC distributions, and hence the design of target surfaces with sharp folds.

## (c) Examples of computed Gaussian curvature

We conclude by demonstrating the use of our general results above, by computing the distribution of concentrated GC along two examples of metric-compatible interfaces between log-spiral LCE patterns, previously described in §3c. We choose one example with a twinned interface (figure 8a) and one with a continuous-metric interface (figure 8d) to illustrate both cases. As a preliminary to both calculations, we note that, in plane polar coordinates, the bend and splay vectors for a log-spiral pattern are simply

$$\mathbf{b} = \mathbf{n}^\perp \sin \frac{\alpha}{r} \quad \text{and} \quad \mathbf{s} = \mathbf{n} \cos \frac{\alpha}{r}. \tag{4.8}$$

For our first example, we take two spirals with $\alpha = \pm\arctan(1/\sqrt{\lambda_\parallel \lambda_\perp})$ in our standard configuration, with a twinned interface along the vertical $y$-axis, as shown in figure 8a. This value of $\alpha$ was chosen as it is the critical value of $\alpha$ between cones and anticones, such that the spirals would remain flat upon individual actuation (figure 5), and thus the only GC in the resultant surface will reside on the interface. To compute the GC, we first gather the various quantities involved in equation (4.5). Along the interface, the radial direction for the left-hand pattern is simply $\mathbf{r}_1 = (c, y)$ which makes an angle $\pi/2 - \arctan(y/c)$ with the $y$-axis, so we may compute $\phi = \xi/2 = \pi/2 - \alpha - \arctan(l/c)$, where we now set $l = y$ as the reference arc-length parameter of the interface. Substituting these values into equation (4.5) along with the bend and splay reveals that the distribution of GC is described by

$$\frac{d\Omega}{dl} = -\frac{\lambda_\parallel \lambda_\perp}{\lambda_\perp^2 \sin^2 \phi + \lambda_\parallel^2 \cos^2 \phi} \frac{2c}{c^2 + l^2} + 2\left( \frac{\lambda_\parallel}{\lambda_\perp} \frac{\sin\alpha \cos\phi}{\sqrt{c^2 + l^2}} + \frac{\lambda_\perp}{\lambda_\parallel} \frac{\cos\alpha \sin\phi}{\sqrt{c^2 + l^2}} \right). \tag{4.9}$$

A plot of this distribution is shown in figure 8b. We see that the GC has a non-trivial distribution that decays away from the origin, but is positive on one side and negative on the other. The corresponding simulation (figure 8c) shows how a surface can accommodate these properties: the crease has the same sense of fold-angle throughout, but changes the finite curvature of the ridge line from positive to negative by tracing out a planar space-curve with a maximum and a minimum. The simulations also reveal that the singular GC above is, in fact, slightly smeared out transverse to the fold, blunting the fold and avoiding the infinite bend energy associated with a truly sharp feature, at the cost of some local stretching energy.

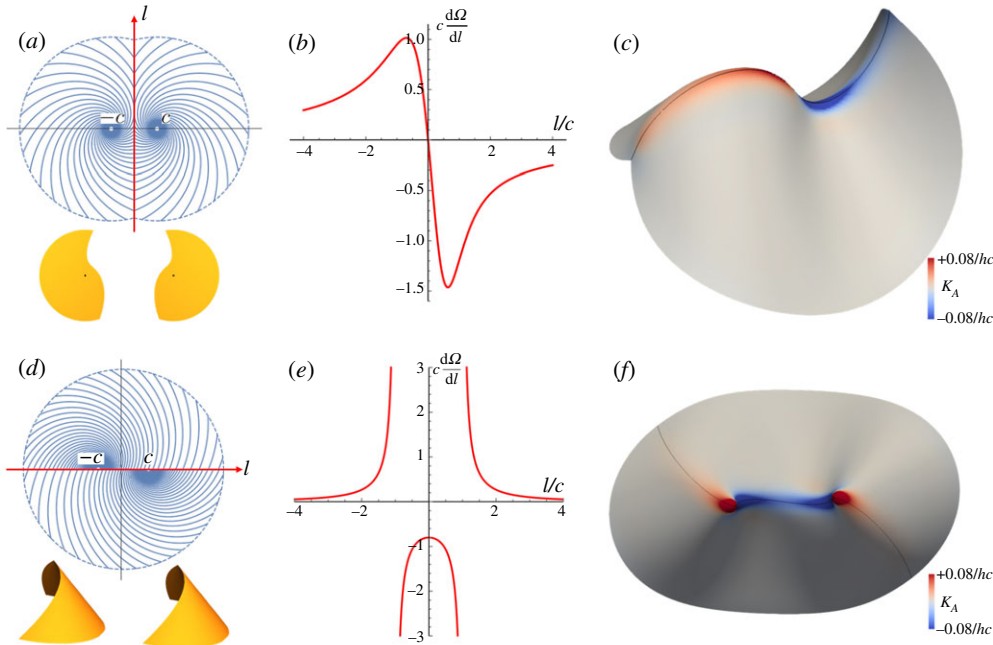

**Figure 8.** (*a–c*) The reference domain ((*a*), top), actuated conical shapes ((*a*), bottom), analytical concentrated GC (*b*), and simulation (*c*) for the twinning interface between two spiral patterns with $\alpha_1 = -\alpha_2 = 1.25$. (*d,e*) The plots of counterparts for the continuous-metric interface between two spiral patterns with $\alpha_1 = \alpha_2 = 1.173$. In our simulations, the thickness *h* is approximately 1% of the sample size. (Online version in colour.)

For our second example, we take spirals with the same angle $\alpha = \alpha_1 = \alpha_2$ in our standard configuration, and joined along the continuous-metric interface along the *x*-axis. Recalling the integrated GC (4.7) for the continuous-metric interface, we have

$$\frac{d\Omega}{dl} = \left( \frac{\lambda_\parallel}{\lambda_\perp} - \frac{\lambda_\perp}{\lambda_\parallel} \right) \frac{c \sin(2\alpha)}{c^2 - l^2}, \tag{4.10}$$

which is plotted in figure 8*e*. In this case, the distribution is also finite near the origin and decays at large distances, but it is negative in the middle, and positive in the extremes. Interestingly, the distribution diverges where the interface passes through the spiral centres, leading to cone-like tips in the actuation surface, which now fall on the interface. Quantifying the GC concentrated at these tips would require a more careful application of Gauss–Bonnet [52] at the point, rather than along lines as done here. Again, the simulated shape shows the same pattern of GC, leading to a surface with a line-like saddle in the middle, joining two positive tips. As previously, the simulated ridges are blunted, avoiding divergent bend.

Overall, these examples illustrate that both types of interface generically bear concentrated GC, and the analytic computation of its distribution gives considerable insight into the form of the actuated surface. In the future, these results may allow the programming of the shapes of such intrinsically curved folds. For example, both simulations in figure 8 show that the region with positive concentrated GC curves downward while the negative region curves upward upon actuation. This is because the transverse principal curvature along the interface does not change its sign, restricted by the domain's boundary, and the longitudinal principal curvature must therefore change sign when the concentrated GC does. This suggests that one can potentially program the concentrated GC along metric-compatible interfaces to obtain an actuated shape with desired directional curving properties.

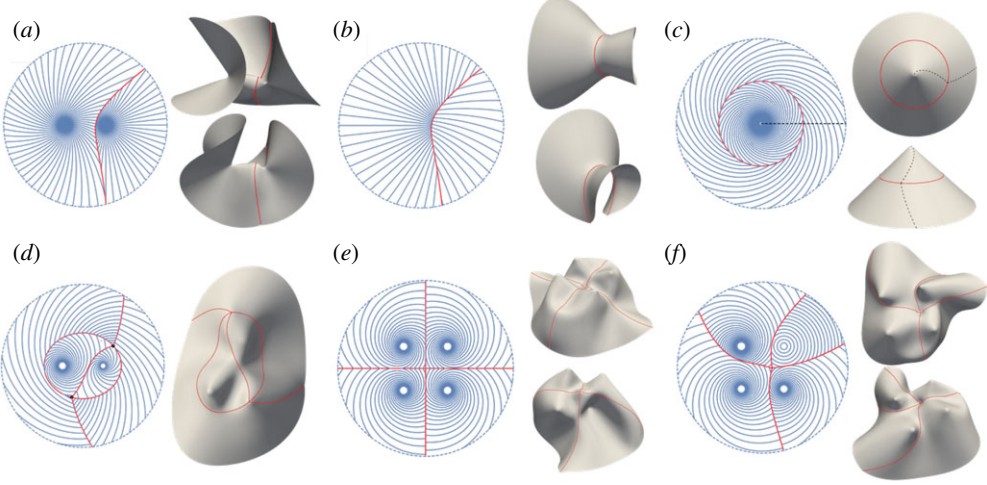

**Figure 9.** (*a*) An interface between two log-spiral patterns that individually actuate to anticones. (*b*) An interface between patterns with no tips by taking the complement of (*a*). (*c*) A circular twinning interface between two concentric spirals with opposite angles ($\alpha_1 = -\alpha_2$). The actuation induces programmable twists. (*d,f*) Complex interfaces and topographies by stitching together patterns with metric-compatible interfaces. (Online version in colour.)

## 5. Discussion

In this paper, we have presented a systematic method for combining different patterns of shape change in a single morphing sheet by stitching them together piecewise via metric-compatible interfaces. This approach enlarges the design space by allowing simple patterns to be used as building blocks for more complex patterns, and also paves the way for multi-material sheets that combine active and passive regions, or even regions with different active materials.

Our study reveals a key difference between active materials such as LCEs that undergo anisotropic local shape changes, and those like swelling gels where the shape change is locally isotropic. Given two patterns of LCE type, there are infinitely many interfaces available along which the metric is discontinuous but *twinned* so that the interface itself is compatible. There may also be a finite number of interfaces in which the metric is continuous across the interface. By contrast, in gel-like systems, only the latter type of interface is available, strongly limiting the possibilities available from stitching. This distinction is seen when combining patterns of the same type of actuation, and also when trying to combine active and passive regions.

To illustrate how this stitching approach dramatically enlarges the design space, we have found all the compatible interfaces between pairs of LCE log-spiral patterns, that would individually actuate to (anti)cones. For every pair of spirals, there are infinitely many twinned interfaces available, generating a rich set of actuated topographies. In our original presentation (figure 6), we demonstrated examples with a single interface that ran between the spiral centres, leading to a surface with two conical tips separated by a curved ridge along the interface. However, the full set of surfaces enabled is much broader than this, as highlighted in figure 9. For example, we may also obtain anti-tips by using spirals that individually make anticones (figure 9*a*), or landscapes with no tips at all by taking the tip-free side (complement) of each pattern (figure 9*b*). A particularly interesting case arises for concentric spirals with opposite angle, which actuates to a simple cone in which the tip and flank twist in opposite senses (figure 9*c*) opening the possibility of designing the twist of a surface, as well as its shape. Moreover, one may also create much more complex patterns with multiple regions connected by multiple compatible interfaces (figure 9*d–f*). Intersections between interfaces generically have an angular surplus or deficit on actuation, generating their own (anti-)tips. The resultant shapes commonly show rich multi-stability, and such patterns may be repeated indefinitely to make switchable textures.

Our study also reveals that the metric-compatible interfaces generically bear concentrated GC, and thus produce sharp folds in the actuated surface. Perhaps surprisingly, such creases

are found for both *twinned* and *continuous* interfaces. Looking ahead, we expect these creases to have interesting and rich mechanical properties, as they are intrinsically curved and cannot be flattened without stretch. For example, ribbon-shaped actuators containing such interfaces will curve into a folded arc on actuation, offering a mode of actuation reminiscent of a conventional bi-layer bender, but with the stretch-based strength of metric mechanics. We reserve for future work a detailed exploration of the mechanics and geometry of such intrinsically curved ridges, which are fundamentally different to the familiar curved folds in isometric origami.

Finally, we note that a key premise of this work has been that stitched interfaces must be metric compatible, otherwise the two regions disagree over the length of the interface on actuation, leading to large internal stresses and ultimately, material failure. However, in the experimental literature, one may find examples of both compatible [2,27,49] and incompatible [9,33,42,57] interfaces. The compatible interfaces follow our interfacial metric mechanics framework exactly, and one can see compatible curved interfaces in the actuated configuration. An interesting middle ground is explored in [42], where a fundamentally compatible interface is biased to actuate up rather than down by the inclusion of a sliver of incompatible material along the interface itself. At the other extreme, the incompatible interfaces cannot be understood in the framework of metric mechanics, as there is no isometry of the metric available. However, it would be very interesting to quantify experimentally the extent to which such incompatible interfaces really are prone to fatigue and failure under actuation, and to formulate a theoretical approach that can predict their resultant shapes and mechanics.

Data accessibility. Electronic supplementary material data are available from the corresponding author upon reasonable request. The software used in this paper can be accessed at https://www.repository.cam.ac.uk/handle/1810/312168 with full permission.

Electronic supplementary material is available online [58].

Authors' contributions. F.F.: investigation, validation, writing—original draft, writing—review and editing; D.D.: investigation, software, validation, writing—original draft, writing—review and editing; M.W.: conceptualization, funding acquisition, supervision; J.S.B.: conceptualization, formal analysis, funding acquisition, investigation, supervision, writing—original draft, writing—review and editing.

All authors gave final approval for publication and agreed to be held accountable for the work performed therein.

Conflict of interest declaration. We declare that we have no competing interests.

Funding. F.F and M.W. were supported by the EPSRC (grant no. EP/P034616/1). D.D. was supported by the EPSRC Centre for Doctoral Training in Computational Methods for Materials Science (grant no. EP/L015552/1). J.S.B. was supported by a UKRI 'future leaders fellowship' (grant no. MR/S017186/1).

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
