## [Peer Review File · Proceedings. Mathematical, Physical, and Engineering Sciences]

Review History

RSPA-2022-0230.R0 (Original submission)

Review form: Referee 1

Is the manuscript an original and important contribution to its field?

Excellent

Is the paper of sufficient general interest?

Excellent

Is the overall quality of the paper suitable?

Excellent

Can the paper be shortened without overall detriment to the main message?

Yes

Do you think some of the material would be more appropriate as an electronic appendix?

No

Do you have any ethical concerns with this paper?

No

Recommendation?

Accept as is

Comments to the Author(s)

This paper provides a mathematical model to study the general condition for the formation of geometrically compatible interfaces in planar soft active sheets, with spatially patterned material properties, undergoing spontaneous deformation under an external stimulus. The paper highlighted the very interesting discrepancy between an LCE system and a gel system, where the former with anisotropic shape morphing has many more possible compatible interfaces when actuated, than the latter with isotropic shape change.

This paper is very well written. The message from the mathematical model is well presented, clear, and highly interesting. The novelty by providing different possibilities to form geometrically compatible interfaces will inspire new experimental discoveries and related applications. Overall I am delighted to read this paper and recommend its acceptance without further revision.

There is only one side question out of curiosity: the authors are clearly familiar with the works of Plucinsky and Bhattacharya on introducing small twists during the director pattern to form a biased pyramid of LCE (e.g., 2018 Soft Matter). What will be the effect of such "imperfections" on the general condition of compatible interfaces? Some "stability analysis" due to such "noise" seems interesting.

Review form: Referee 2**Is the manuscript an original and important contribution to its field?**

Excellent

Is the paper of sufficient general interest?

Excellent

Is the overall quality of the paper suitable?

Excellent

Can the paper be shortened without overall detriment to the main message?

Yes

Do you think some of the material would be more appropriate as an electronic appendix?

No

Do you have any ethical concerns with this paper?

No

Recommendation?

Accept with minor revision (please list in comments)

Comments to the Author(s)

This paper studies the flat active sheet programmed with different patterns in different regions. Once actuated, the flat sheet will morph into a curved surface. Specifically, the authors focus on the interface between regions with different patterns. The authors describe the geometrically compatible condition of the interface and applies the general rule to materials with different active modes, including LCE-like, gel-like and active/passive materials. For LCE-like materials,

authors demonstrate several designs of compatible patterns. The authors also discussed actuated surfaces with concentrated Gaussian curvature of LCE-like sheet. The problem studied in this work is surely of interest to community of active materials. I suggest the acceptance of this manuscript after minor revision.

Some specific questions are as follows:

- (1) The discussion in the paper could be applied to certain kinds of active mode instead of certain active materials, like gel and LCE mentioned in the paper. The LCE shows anisotropic contraction and expansion, while the gel usually exhibits both expansion in x and y directions. Could the theory work for both LCEs and gels? I suggest discussing their similarities and differences with LCEs and swelling gels in the manuscript.
- (2) It could be better to give some comparison between theory and experiment (from literature) of a particular active material.
- (3) In section 3, computational method in shell energy is introduced. The authors should elaborate more on the energy formula. And a and b in Eq. S1 should be given explicitly to make the paper self-contained.
- (4) The authors may elaborate a bit more on the prefactors of the stretching and bending energy. Is it a ratio of the penalty between stretching and bending? If yes, the sheet in Fig 3d may not always form a continuous surface even in simulation.
- (5) The out-of-plane shape changes within a patterned domain can be suppressed by increasing the sheet thickness. In the current model, the compatible stitched interface is achieved by designing the patterns around it. Is it possible to suppress the incompatible interface by increasing the sheet thickness?

Review form: Referee 3

Is the manuscript an original and important contribution to its field?

Good

Is the paper of sufficient general interest?

Good

Is the overall quality of the paper suitable?

Good

Do you have any ethical concerns with this paper?

No

Recommendation?

Accept with minor revision (please list in comments)

Comments to the Author(s)

See attached.

Decision letter (RSPA-2022-0230.R0)

23-May-2022

Dear Dr Feng,

On behalf of the Editor, I am pleased to inform you that your Manuscript RSPA-2022-0230 entitled "Interfacial metric mechanics: stitching patterns of shape change in active sheets" has been accepted for publication subject to minor revisions in Proceedings A. Please find the referees' comments below.

The reviewer(s) have recommended publication, but also suggest some minor revisions to your manuscript. Therefore, I invite you to respond to the reviewer(s)' comments and revise your manuscript. Please note that we have a strict upper limit of 28 pages for each paper. Please endeavour to incorporate any revisions while keeping the paper within journal limits. Please note that page charges are made on all papers longer than 20 pages. If you cannot pay these charges you must reduce your paper to 20 pages before submitting your revision. Your paper has been ESTIMATED to be 19 pages. We cannot proceed with typesetting your paper without your agreement to meet page charges in full should the paper exceed 20 pages when typeset. If you have any questions, please do get in touch.

It is a condition of publication that you submit the revised version of your manuscript within 7 days. If you do not think you will be able to meet this date please let me know in advance of the due date.

To revise your manuscript, log into <https://mc.manuscriptcentral.com/prsa> and enter your Author Centre, where you will find your manuscript title listed under "Manuscripts with Decisions." Under "Actions," click on "Create a Revision." Your manuscript number has been appended to denote a revision.

You will be unable to make your revisions on the originally submitted version of the manuscript. Instead, revise your manuscript and upload a new version through your Author Centre.

When submitting your revised manuscript, you will be able to respond to the comments made by the referee(s) and upload a file "Response to Referees" in Step 1: "View and Respond to Decision Letter". Please provide a point-by-point response to the comments raised by the reviewers and the editor(s). A thorough response to these points will help us to assess your revision quickly. You can also upload a 'tracked changes' version either as part of the 'Response to reviews' or as a 'Main document'.

IMPORTANT: Your original files are available to you when you upload your revised manuscript. Please delete any redundant files before completing the submission process.

When uploading your revised files, please make sure that you include the following as we cannot proceed without these:

- 1) A text file of the manuscript (doc, txt, rtf or tex), including the references, tables (including captions) and figure captions. Please remove any tracked changes from the text before submission. PDF files are not an accepted format for the "Main Document".
- 2) A separate electronic file of each figure (tif, eps or print-quality pdf preferred). The format should be produced directly from original creation package, or original software format.
- 3) Electronic Supplementary Material (ESM): all supplementary materials accompanying an accepted article will be treated as in their final form. Note that the Royal Society will not edit or typeset supplementary material and it will be hosted as provided. Please ensure that the supplementary material includes the paper details where possible (authors, article title, journal name). Supplementary files will be published alongside the paper on the journal website and posted on the online figshare repository (<https://figshare.com>). The heading and legend

provided for each supplementary file during the submission process will be used to create the figshare page, so please ensure these are accurate and informative so that your files can be found in searches. Files on figshare will be made available approximately one week before the accompanying article so that the supplementary material can be attributed a unique DOI. Alternatively you may upload a zip folder containing all source files for your manuscript as described above with a PDF as your "Main Document". This should be the full paper as it appears when compiled from the individual files supplied in the zip folder.

Article Funder

Please ensure you fill in the Article Funder question on page 2 to ensure the correct data is collected for FundRef (<http://www.crossref.org/fundref/>).

Media summary

Please ensure you include a short non-technical summary (up to 100 words) of the key findings/importance of your paper. This will be used for to promote your work and marketing purposes (e.g. press releases). The summary should be prepared using the following guidelines:

- *Write simple English: this is intended for the general public. Please explain any essential technical terms in a short and simple manner.
- *Describe (a) the study (b) its key findings and (c) its implications.
- *State why this work is newsworthy, be concise and do not overstate (true 'breakthroughs' are a rarity).
- *Ensure that you include valid contact details for the lead author (institutional address, email address, telephone number).

Cover images

We welcome submissions of images for possible use on the cover of Proceedings A. Images should be square in dimension and please ensure that you obtain all relevant copyright permissions before submitting the image to us. If you would like to submit an image for consideration please send your image to proceedingsa@royalsociety.org

Open Access

You are invited to opt for open access, our author pays publishing model. Payment of open access fees will enable your article to be made freely available via the Royal Society website as soon as it is ready for publication. For more information about open access please visit <https://royalsociety.org/journals/authors/open-access/>. The open access fee for this journal is £1700/\$2380/€2040 per article. VAT will be charged where applicable. Please note that if the corresponding author is at an institution that is part of a Read and Publishing deal you are required to select this option. See <https://royalsociety.org/journals/librarians/purchasing/read-and-publish/read-publish-agreements/> for further details.

Once again, thank you for submitting your manuscript to Proceedings A and I look forward to receiving your revision. If you have any questions at all, please do not hesitate to get in touch.

Best wishes
Raminder Shergill
proceedingsa@royalsociety.org
Proceedings A

on behalf of
 Professor Yihui Zhang
 Board Member
 Proceedings A

Reviewer(s)' Comments to Author:

Referee: 1

Comments to the Author(s)

This paper provides a mathematical model to study the general condition for the formation of geometrically compatible interfaces in planar soft active sheets, with spatially patterned material properties, undergoing spontaneous deformation under an external stimulus. The paper highlighted the very interesting discrepancy between an LCE system and a gel system, where the former with anisotropic shape morphing has many more possible compatible interfaces when actuated, than the latter with isotropic shape change.

This paper is very well written. The message from the mathematical model is well presented, clear, and highly interesting. The novelty by providing different possibilities to form geometrically compatible interfaces will inspire new experimental discoveries and related applications. Overall I am delighted to read this paper and recommend its acceptance without further revision.

There is only one side question out of curiosity: the authors are clearly familiar with the works of Plucinsky and Bhattacharya on introducing small twists during the director pattern to form a biased pyramid of LCE (e.g., 2018 Soft Matter). What will be the effect of such "imperfections" on the general condition of compatible interfaces? Some "stability analysis" due to such "noise" seems interesting.

Referee: 2

Comments to the Author(s)

This paper studies the flat active sheet programmed with different patterns in different regions. Once actuated, the flat sheet will morph into a curved surface. Specifically, the authors focus on the interface between regions with different patterns. The authors describe the geometrically compatible condition of the interface and applies the general rule to materials with different active modes, including LCE-like, gel-like and active/passive materials. For LCE-like materials, authors demonstrate several designs of compatible patterns. The authors also discussed actuated surfaces with concentrated Gaussian curvature of LCE-like sheet. The problem studied in this work is surely of interest to community of active materials. I suggest the acceptance of this manuscript after minor revision.

Some specific questions are as follows:

- (1) The discussion in the paper could be applied to certain kinds of active mode instead of certain active materials, like gel and LCE mentioned in the paper. The LCE shows anisotropic contraction and expansion, while the gel usually exhibits both expansion in x and y directions. Could the theory work for both LCEs and gels? I suggest discussing their similarities and differences with LCEs and swelling gels in the manuscript.
- (2) It could be better to give some comparison between theory and experiment (from literature) of a particular active material.
- (3) In section 3, computational method in shell energy is introduced. The authors should elaborate more on the energy formula. And a and b in Eq. S1 should be given explicitly to make the paper self-contained.
- (4) The authors may elaborate a bit more on the prefactors of the stretching and bending energy. Is it a ratio of the penalty between stretching and bending? If yes, the sheet in Fig 3d may not always form a continuous surface even in simulation.
- (5) The out-of-plane shape changes within a patterned domain can be suppressed by increasing the sheet thickness. In the current model, the compatible stitched interface is achieved by

designing the patterns around it. Is it possible to suppress the incompatible interface by increasing the sheet thickness?

Referee: 3

Comments to the Author(s)

See attached.

Author's Response to Decision Letter for (RSPA-2022-0230.R0)

See Appendix A.

RSPA-2022-0230.R1 (Revision)

Review form: Referee 2

Is the manuscript an original and important contribution to its field?

Excellent

Is the paper of sufficient general interest?

Excellent

Is the overall quality of the paper suitable?

Excellent

Do you have any ethical concerns with this paper?

No

Recommendation?

Accept as is

Comments to the Author(s)

Very nice work.

Review form: Referee 3

Is the manuscript an original and important contribution to its field?

Good

Is the paper of sufficient general interest?

Good

Is the overall quality of the paper suitable?

Excellent

Do you have any ethical concerns with this paper?

No

Recommendation?

Accept as is

Comments to the Author(s)

The authors have addressed my comments. I think it is a nice paper!

Decision letter (RSPA-2022-0230.R1)

09-Jun-2022

Dear Dr Feng

I am pleased to inform you that your manuscript entitled "Interfacial metric mechanics: stitching patterns of shape change in active sheets" has been accepted in its final form for publication in Proceedings A.

Our Production Office will be in contact with you in due course. You can expect to receive a proof of your article soon. Please contact the office to let us know if you are likely to be away from e-mail in the near future. If you do not notify us and comments are not received within 5 days of sending the proof, we may publish the paper as it stands.

As a reminder, you have provided the following 'Data accessibility statement' (if applicable). Please remember to make any data sets live prior to publication, and update any links as needed when you receive a proof to check. It is good practice to also add data sets to your reference list. *Statement (if applicable):* The software used in this paper can be accessed at <https://www.repository.cam.ac.uk/handle/1810/312168> with full permission.

Under the terms of our licence to publish you may post the author generated postprint (ie. your accepted version not the final typeset version) of your manuscript at any time and this can be made freely available. Postprints can be deposited on a personal or institutional website, or a recognised server/repository. Please note however, that the reporting of postprints is subject to a media embargo, and that the status the manuscript should be made clear. Upon publication of the definitive version on the publisher's site, full details and a link should be added.

You can cite the article in advance of publication using its DOI. The DOI will take the form: 10.1098/rspa.XXXX.YYYY, where XXXX and YYYY are the last 8 digits of your manuscript number (eg. if your manuscript number is RSPA-2017-1234 the DOI would be 10.1098/rspa.2017.1234).

For tips on promoting your accepted paper see our blog post: <https://royalsociety.org/blog/2020/07/promoting-your-latest-paper-and-tracking-your-results/>

On behalf of the Editor of Proceedings A, we look forward to your continued contributions to the Journal.

Sincerely,
Raminder Shergill

proceedingsa@royalsociety.org

on behalf of
Professor Yihui Zhang
Board Member
Proceedings A

Reviewer(s)' Comments to Author:

Referee: 3

Comments to the Author(s)

The authors have addressed my comments. I think it is a nice paper!

Referee: 2

Comments to the Author(s)

Very nice work.

Appendix A

Response to reviewers: Interfacial metric mechanics: stitching patterns of shape change in active sheets.

Fan Feng¹, Daniel Duffy¹, Mark Warner², and John S. Biggins¹

¹Department of Engineering, University of Cambridge, Cambridge CB2 1PZ, United Kingdom

²Department of Physics, University of Cambridge, Cambridge CB3 0HE, United Kingdom

May 30, 2022

We thank all the referees for their uniformly positive assessment of our paper, and also for their constructive comments that helped us to improve it further. We provide point-to-point responses (highlighted in blue) to the referees' comments as follows. The revised text of the manuscript is highlighted in red in the file "Main_revision_highlighted.pdf".

Referee 1.

Comments to the Author(s): This paper provides a mathematical model to study the general condition for the formation of geometrically compatible interfaces in planar soft active sheets, with spatially patterned material properties, undergoing spontaneous deformation under an external stimulus. The paper highlighted the very interesting discrepancy between an LCE system and a gel system, where the former with anisotropic shape morphing has many more possible compatible interfaces when actuated, than the latter with isotropic shape change. This paper is very well written. The message from the mathematical model is well presented, clear, and highly interesting. The novelty by providing different possibilities to form geometrically compatible interfaces will inspire new experimental discoveries and related applications. Overall I am delighted to read this paper and recommend its acceptance without further revision.

We thank Referee 1 for this entirely positive assessment and the recommendation for publication.

There is only one side question out of curiosity: the authors are clearly familiar with the works of Plucinsky and Bhattacharya on introducing small twists during the director pattern to form a biased pyramid of LCE (e.g., 2018 Soft Matter). What will be the effect of such "imperfections" on the general condition of compatible interfaces? Some "stability analysis" due to such "noise" seems interesting.

In these boundaries the boundary is chosen to be compatible, as in the cases treated in our paper. However, along the boundary itself, a sliver of incompatible material is added later in the design process to bias the actuation. We agree that what happens in this situation is an interesting question, and not resolved in the present work. We speculate that the answer depends on the width of the sliver compared to sheet thickness, and also the emergent mechanical length scale of the curvature of the fold: if the width is large, we would expect (hierarchical?) wrinkling to

accommodate the incompatibility, while if it is short, a 3D elastic treatment is required rather than a shell/plate theory. We have added a comment to the final paragraph of the paper mentioning such interfaces, but make no claim to predict how they will behave.

Referee 2. This paper studies the flat active sheet programmed with different patterns in different regions. Once actuated, the flat sheet will morph into a curved surface. Specifically, the authors focus on the interface between regions with different patterns. The authors describe the geometrically compatible condition of the interface and applies the general rule to materials with different active modes, including LCE-like, gel-like and active/passive materials. For LCE-like materials, authors demonstrate several designs of compatible patterns. The authors also discussed actuated surfaces with concentrated Gaussian curvature of LCE-like sheet. The problem studied in this work is surely of interest to community of active materials. I suggest the acceptance of this manuscript after minor revision.

We thank Referee 2 for the positive comment and the recommendation for publication.

Some specific questions are as follows:

(1) The discussion in the paper could be applied to certain kinds of active mode instead of certain active materials, like gel and LCE mentioned in the paper. The LCE shows anisotropic contraction and expansion, while the gel usually exhibits both expansion in x and y directions. Could the theory work for both LCEs and gels? I suggest discussing their similarities and differences with LCEs and swelling gels in the manuscript.

We are confused by this comment. Our discussion in Section 2(a) use on a general mathematical model of a spontaneous deformation, and is not limited to particular materials or types of modes. However, we already have discussed the specific cases of LCEs and gels in Section 2(b), highlighting, in the words of referee 1 “the very interesting discrepancy between an LCE system and a gel systems” - this appears to be exactly what the referee is asking for.

(2) It could be better to give some comparison between theory and experiment (from literature) of a particular active material.

We found this a very useful comment. We have added some comparison to the discussion section, including an example of arrays of cones [T. Guin et al., Nat. Commun., 9(1):1–7, 2018] and an example of lines of Gaussian curvature [D. Duffy et al., J. Appl. Phys., 129(22):224701, 2021]. We hope this clarifies that such interfaces are already a topic of active experimental research, and will allow the reader to explore the experimental systems if they so desire.

(3) In section 3, computational method in shell energy is introduced. The authors should elaborate more on the energy formula. And a and b in Eq. S1 should be given explicitly to make the paper self-contained.

We agree that it is a good idea to include some more details about the shell energy that underpins our simulation. We have therefore elaborated a little in the supplement, including giving definitions of the first and second fundamental forms (a and b), and discussing the physical origin of the energy. We have also added a reference to the supplement from the main text so that the interested reader can find these details. We do still think that these details belong in a supplement, as they are not new, and are ancillary to our main content which is focused on geometry rather than mechanics.

(4) The authors may elaborate a bit more on the prefactors of the stretching and bending energy. Is it a ratio of the penalty between stretching and bending? If yes, the sheet in Fig 3d may not always form a continuous surface even in simulation.

Yes, the h and h^3 prefactors in the energy represent the penalty for stretching and bending, as ubiquitous in shell and plate mechanics. We have clarified this origin in our extended explanation

of the shell energy.

With regards to the reviewers question about continuity, in our simulation, we assume the sheet is always in the elastic regime, and no plasticity or fracture occurs. Therefore, the simulated surface is always continuous. We suspect the referee actually meant sharp (i.e. discontinuous gradient) rather than a discontinuous displacement. In fact, we see that the interfaces are always blunted by a stretch-bend trade-off, and only become sharp only in the limit of zero thickness. We intend to study these mechanical considerations in a future work.

(5) The out-of-plane shape changes within a patterned domain can be suppressed by increasing the sheet thickness. In the current model, the compatible stitched interface is achieved by designing the patterns around it. Is it possible to suppress the incompatible interface by increasing the sheet thickness?

This is a very interesting question, but one which we largely reserve for future work: our emphasis here is on the geometry of thin isometric sheets, not the mechanical stretch/bend trade offs that emerge in thick ones. We believe the actuated interface in a physical system is somewhat blunted by such mechanics, and would become progressively smoother as the thickness increases. There may well be a critical thickness past which the sheet remains planar despite having a non-flat metric, but at this point it cannot be regarded as thin or isometric.

Referee 3.

This paper describes how to stitch together so-called active patterns at an interface. On actuation each such pattern has a different induced metric that constrains how the pattern morphs into a 3D configuration. Stitching together two such patterns, thus, requires that certain rank-one compatibility conditions be satisfied at the patterned interface prior to actuation. To begin, the authors characterize all possible compatible interfaces for all possible induced metrics. They then show how this characterization furnishes design rules to stitch together two active patterns. They do so in the context of well-known active sheet examples: Two swelling gels, an active and passive region, and, most thoroughly, two LCE patterns. Although this paper is primarily theoretical, the examples combine the design rules with numerical simulations in a way that opens the door for a lot of further exploration with experimental groups. Lastly, the authors develop rather elegant and general formulas for calculating the Gaussian curvature of an interface in these systems. This paper is clear, concise and authoritative. It asks and resolves interesting questions in mechanic about the nature of active sheet embeddings. The ideas also have the potential to inform the design of active sheets, which is an important emerging topic in materials science. The paper, in my opinion, is suitable for publication in PRSA, after the authors address my comments below.

We thank Referee 3 for the careful reading, constructive suggestions and encouraging comments.

Major comment.

- Metric compatibility is a necessary conditions. I think more needs to be conveyed about the fact that you are characterizing a necessary condition for an embedding, not the necessary and sufficient conditions. You do so with the example in Fig 3d, but it feels kind of “swept under the rug” in my opinion. As I understand it, your hope is that there are a rich enough family of isometric deformations on the two sides of an interface such that a metric compatible interface implies an actual embedding. I basically think you should be upfront and transparent about this “hope” from the start. There does seem to be a reasonable sales pitch for this view, albeit it requires a thoughtful paragraph: In terms of design, metric compatibility is the only designable feature of the characterization of interfaces anyway. You also have access to simulations, which can test the validity of the designs obtained by your characterization, ect., etc. . . This feels like a discussion

that needs to be had either at the beginning or end of Section 2(b).

We are pleased to add some discussion of this very interesting point, as the referee suggests. We agree that metric compatibility is a necessary condition, rather than a sufficient condition for an embedding. We are unaware of any embedding theorems that can definitively settle the question of sufficiency: for example, Nash embedding assumes smooth embeddings and smooth metrics. However, it is our numerical experience that well behaved embeddings with only a sharp ridge along the interface are generically available. We also agree with the referee's point that metric compatibility is the only intrinsic design tool available. We have elaborated on these points at the start of Section 2(b), as suggested.

Minor comments.

- Page 3, line 36: The phrasing “Even more relatedly” sounds awkward to me and is vague. I'd suggest something like “Beyond describing the physics of interfaces, . . .”.

We have done so.

- Page 4 line 23: Given your notation in this work, I think you should replace the identity I with \mathbf{I} or \mathbf{I} or \mathbf{Id} for consistency.

We have used \mathbf{I} to represent identity throughout the paper.

- Eq (2.3): The implied notation $\mathbf{a}_i = \mathbf{U}_i^2$ might be “too slick” for a typical reader. I think you should define it explicitly.

We have defined \mathbf{a} explicitly below eq. (2.2) and pointed \mathbf{a}_i to it.

- Page 4, line 53: “One may solve algebraically, but a graphical approach is highly instructive for understanding when these different cases arise, which will transpire to determine the ease of constructing such stitched interfaces between patterns.” I feel like this sentence needs a re-organization. Maybe something like: “One may characterize the equation algebraically, but a graphical approach is highly instructive for understanding when these different cases arise and the relative ease of constructing such stitched interfaces between patterns.”

We have changed it to “One may characterize the equation algebraically, but a graphical approach is highly instructive for understanding these different cases and constructing such stitched interfaces between patterns.”

- For case (ii) on page 5, I'm confused. You have a critical angle. At this angle, there is only one possible interface? Even though this “1-solution” case is not generic, shouldn't it still be clearly stated in your characterization?

Yes, the critical angle corresponds to one possible interface. We have stated this more clearly.

- Page 6, line 58: “symmetrically” sounds awkward here. I guess the correct thing to say is something like “the actuations $\mathbf{U}_{1,2}(\mathbf{x})$ are mirrors of each other across the interface”.

We have changed it to “We call interfaces constructed along these directions *twinning interfaces*, as the directors are discontinuous at the interface but symmetrically satisfy the twinning condition $|\mathbf{n}_1 \cdot \mathbf{t}| = |\mathbf{n}_2 \cdot \mathbf{t}|$. Then the the deformations $\mathbf{U}_{1,2}(\mathbf{x})$ are mirrors of each other across the interface.”

- Eq (4.2): Why is \mathbf{l}_A boldface in the integration domain, and not boldface in the text?

We use boldface \mathbf{l}_A to represent a curve, and l_A to represent its arclength. We have unified the notations.

- Eq (4.3): This is a very minor comment. You can save some space by simply replacing $(\nabla \times \mathbf{n})$ with $\nabla \cdot \mathbf{n}_\perp$. Also, in solid mechanics, the 2D curl is not used all that much. So I personally stumbled over $\nabla \times \mathbf{n}$ for a bit thinking that $(\nabla \times \mathbf{n})\mathbf{n}_\perp$ was a tensor, not a vector.

We have pointed out that the curl is taken in a 2D scalar sense in the original version. We prefer keeping this form to make the notation consistent with the related references.

- Page 13: I think you should be a bit more mathematically accurate about your integrals. I sympathize with the attempt to keep a concise notation, but, as you well know, it is not proper to have $\mathbf{x}(l)$ indicate the domain of integration, while also integrating over l . In addition, $\mathbf{x}(l)$ indicates both a point on a curve and the curve itself. My feeling is that there is too much “abuse of notation” here that could confuse a typical reader.

We have changed the domain of integration to $[0, \bar{l}]$ where \bar{l} is the total arclength of the curve.

- Eq (4.5): I think you should pick one of the \mathbf{a} ’s in the integrand, rather than use the implied notation $\sqrt{\mathbf{t} \cdot \mathbf{a}_i \cdot \mathbf{t}} = \sqrt{\mathbf{t} \cdot \mathbf{a} \cdot \mathbf{t}}$. Maybe write $dl_A/dl = \sqrt{\mathbf{t} \cdot \mathbf{a}_1 \cdot \mathbf{t}} = \sqrt{\mathbf{t} \cdot \mathbf{a}_2 \cdot \mathbf{t}}$ as a reminder to the reader.

We have done so.

- Page 14, line 6: interface should be interfaces.

We have corrected it.